

# What even is 'Climate'?

Oliver Bothe[1]

[1]Helmholtz-Zentrum Geesthacht, Institute of Coastal Research, Max-Planck-Strasse 1, 21502, Geesthacht, Germany

**Correspondence:** Oliver Bothe (ol.bothe@gmail.com, oliver.bothe@hzg.de)

**Abstract.** Although the concept of climate is easy to understand, there is not any uncontroversial definition of it. Most definitions fall back to the simple formulation that 'climate is the statistics of weather'. Recent attempts at a definition called versions of this saying vague. Climate is policy-relevant, and discussions on climate and climate change benefit from clarity on the topic. Beyond the policy relevance a definition should also be valid for scientific purposes and for individual views. It
has to account for a general concept and individual instances of climate. Here, I try to highlight why the flexibility and the immediacy of the colloquial definition fit the topic. This defence shifts the lack of a clear definition towards the term 'weather' and the time-scales separating weather and climate.

## 1  Introduction

"Climate is what you expect, weather is what you get" (e.g., Geographical Association and London Geographical Institute,
1902; Lorenz, unpublished). Or: "Climate is the statistics of weather" (e.g., McBean et al., 1992; Easterling et al., 1999; Farmer, 2014; Molua and Lambi, 2007; von Storch, 2004; Roe and O'Neal, 2009; Roe, 2009). Public discourse and scientific literature frequently use versions of these colloquial sayings.

It is easy to agree with such statements as climate and weather have meaning for most people. Everybody experiences weather and most have, likely, a grasp on the notion of different climates. This is despite that the concepts of 'climate' and
'weather' developed over centuries in different cultural contexts (Heymann, 2010; Stehr and von Storch, 2000, 1998; Barnes, 1921). The references mentioned above all use variations of the colloquial saying and may refer to different aspects of climate. Nevertheless most people will understand the intentions already without a more thorough definition.

Indeed, some may see the question of how to best or correctly define 'climate' as purely academic or philosophic. However, climate is a topic of policies, and discussions on policy benefit from clarity of the topic. Werndl (2015) points to the problems
that may arise from a lack of clarity in talking about climate. Depending on their definition of 'climate', correspondents may come to differing conclusions about the seriousness of climatic changes. The thing called 'climate' in a conversation even may not be climate at all. On the other hand, we use the term 'climate' in various contexts that refer to different temporal and/or spatial scales. 'Climate' has colloquial, scientific, philosophical, and political meanings. Already in this introduction, the use of 'climate' and climate may confuse. One refers to the word and the concept, the other to its realisation.



We can define 'climate' or clarify what we generally mean by 'climate' from any of the perspectives mentioned. We can do this conceptually rigorously or with a view on everyday applications or with a focus on the science of the earth system. Here, I try to consider the term 'climate' and its common definition from a more or less everyday-perspective.

## 1.1 Definitions of climate

In this everyday sense, a definition of 'climate' has to account for scientific and policy purposes but also for individual views.
It should allow for classical climate classification (Köppen, 2011; Farmer and Cook, 2013; Willett, 1931; Barry, 2013) and modern perspectives and the development from the former to the latter (Heymann, 2009).

It is important to distinguish between the template definition of 'climate', which represents the concept, and the instances of climate, that people deal with. The colloquial formulation represents the concept validly but springs from individuals' views of their specific instances. Generally, public discussions mostly deal with specific instances. Therefore one needs to rigorously
specify the instance of current interest.

Naturally one turns to the International Panel on Climate Change (IPCC) for a definition of 'climate'. Their fifth Assessment Report (IPCC, 2013) takes the view that

> "Climate in a narrow sense is usually defined as the average weather, or more rigorously, as the statistical description in terms of the mean and variability of relevant quantities over a period of time ranging from months to
> thousands or millions of years. The classical period for averaging these variables is 30 years, as defined by the World Meteorological Organization. The relevant quantities are most often surface variables such as temperature, precipitation and wind. Climate in a wider sense is the state, including a statistical description, of the climate system."

Climate in both senses are instances of a common template. The American Meteorological Society (AMS) (2016b) describes
climate in a similar way (last modified in the year 2012) as

> "the slowly varying aspects of the atmosphere–hydrosphere–land surface system. It is typically characterized in terms of suitable averages of the climate system over periods of a month or more, taking into consideration the variability in time of these averaged quantities..."

As a sidenote concerning the observable climate, Lovejoy and colleagues suggest to replace the dichotomy in terms of temporal variability between weather and climate with a weather-macroweather-climate trichotomy (e.g., Lovejoy, 2013; Lovejoy and Schertzer, 2013; Lovejoy et al., 2013). The expected state is the macroweather and climate is its evolution at longer time-scales. Combining this with the AMS definition, AMS-climate becomes Lovejoy-macroweather, and Lovejoy-climate is the AMS's "variability in time of these averaged quantities".

Many authors presented their views on the concept of 'climate'. Todorov noted in 1986 a lack of "agreement among cli-
matologists on the definition of the term climate" (Todorov, 1986). A similar notion motivated Bryson's essay on climatology (Bryson, 1997). He distinguishes between Climate (upper-case) and climate (lower-case). In his definition Climate is "the





thermodynamic/hydrodynamic status of the global boundary conditions that determine the concurrent array of weather patterns." Lower-case climate is "the statistical characteristics of the weather assemblage at various places, or "typical weather" ". Lower-case climate agrees with the common dictum. The opening of Von Storch and Zwiers (2001) emphasizes the origins of climatology by stating

> "Climatology was originally a sub-discipline of geography, and . . . [d]escription of the climate consisted primarily
> of estimates of its mean state and estimates of its variability about that state. . . The paradigm of climate research
> 5    evolved . . . towards an understanding of the dynamics of climate. . . Statistics plays an important role in this new
> paradigm."

According to them, 'climate' is a description of the climate system in terms of statistics as also the IPCC writes (IPCC, 2013). The descriptive climatology and the physical and chemical concepts of the atmospheric sciences may provide different perpectives for a definition.

Heymann (2009) details how the term 'climate' and the methods of scientists studying it changed from a descriptive approach to a physical and mathematical one. This relates to the point of Hulme et al. (2009) how climate and expectations of it are not purely scientific but also social constructs. Similarly, Heymann (2010) emphasizes the cultural contexts of different views on climate. A concept of 'climate' has to allow for all these culturally conditioned instances. A definition of the observable climate may be easy, but it is only one instance.

The idea of a culturally constructed view on 'climate' refers on the one hand to the fact, that scientists' understanding of the world influences their view of what they consider to be climate. There are the perspectives of greek philosophers; for example, Ptolemy took a solely astronomical perspective on climate (Frisinger and Frisinger, 1973). There was also Humboldt, who viewed 'climate' as a geographically localised property and as something that directly affects humans (Heymann, 2009). The view on 'climate' developed towards a more dynamical understanding in the middle of the 20th century. Throughout history,
new observational techniques resulted in different constructs of 'climate' (Heymann, 2010).

On the other hand, any societal actor may use a definition of 'climate' that possibly differs from a scholar's. Stehr and von Storch (1995) mention the example of famine in 14th century England that the church interpreted as climatic change and that people's atonement could revert. Societies define their own view on climate. Following Hulme (2009), geographic locations, cultures, and traditions influence individual and collective definitions of 'climate'.

If a member of the public asks climate scientists about the implications of climate change they refer to their specific notions of the current climate. These will differ from the notions of their neighbour as both notions will differ from an "official" estimate of the climate. All three estimates possibly align with the concept. The rigor in the discussion has to apply to the agreement of the discussants on the instance under consideration. These possible ambiguities in talking about 'climate' highlight the need for clear statements what one means when one uses the term 'climate'.

Coming back to the common saying that "climate is the statistics of weather", Werndl (2015) calls the IPCC-version (IPCC, 2013) of this dictum vague. This perceived vagueness and the lack of an agreed on definition led her to systematically approach a rigorous definition. She formulates five conditions on a definition of 'climate':



- There have to be empirical applications, e.g., if there are observations they have to provide the climate.

- If time-periods differ uncontroversially in their climate, the definition has to account for this.

- The definition should apply to climates from the past to the future including the present.

- The definition has to be mathematically well-defined.

5 - She also includes the following desired property of a definition: climate should not depend on our knowledge, i.e. saying what the climate was and whether it will change should be independent of the quality of our knowledge.

In turn, Werndl (2015) and Frigg et al. (2015) arrive at preferred definitions which state that 'climate' is the distribution of climate variables "over time for regimes of varying external conditions". This is similar to Bryson's view (Bryson, 1997). These authors and further colleagues still conclude in Bradley et al. (forthcoming) that "defining climate is nontrivial and there 10 is no generally accepted or uncontroversial definition of climate".

I would add a sixth 'desideratum' which may be incompatible with Werndl's. The definition has to be continuously applicable. If I take overlapping time periods, I obtain different instances of climate. These may be virtually identical or they may differ according to accepted criteria. 'Climate' evolves gradually.

As different as the above descriptions of climate are they do not generally contradict the view that "climate is the statistics of 15 weather". Some see the dictum as vague and unspecific (Werndl, 2015; Frigg et al., 2015), I would call it intuitive, simple, and most importantly flexible. It allows for the variety of instances of the concept which scientists and non-scientists commonly call 'climate'.

By shortly discussing what one may mean by 'weather' and reflecting on the components of the classical notions of climate, I am going to make the point that "Climate as the statistics of weather" is an appropriate definition of the concept of 'climate'. 20 Furthermore, it is usually clear enough as a definition. Problems mainly arises in stating the instance of climate one speaks about.

Bryson (1997), the IPCC (2013), the AMS (American Meteorological Society, 2016b), Storch and von Zwiers (2001), all present slight variations on the colloquial view on climate. These emphasize Todorov's (1986) point that everybody talks about climate but there remains ambiguity about the term. If we for the moment take 'climate' as a term that subsumes information 25 of weather, we need to first consider the term 'weather'.

## 2 On 'weather'

Weather is easier to describe than climate and thus also better defined. However, as hopefully becomes clear in the following, it is a major problem in defending the colloquial definition of 'climate'. My defence may stretch the definition of weather or shift the gap in the definitions towards the term weather.



30   What is weather? How do I experience weather personally right now? It is grey, wet and cold but calm. Temperature is relatively low, it drizzles and there is a closed cloud cover but no wind. Yesterday's weather was similar, but there was more wind and more rain.

If I were on a research vessel, I would also be interested in the wave height. Is the wave height weather?

## 2.1   What is weather?

There is the term 'ocean weather'. There is also the science of space weather which refers mainly to the conditions and
influences of the Sun, the solar wind, and similar contributors on the conditions in the solar system. It often focusses on the space surrounding the Earth (Bothmer and Daglis, 2007; Moldwin, 2008; Baker, 2002).

Geology uses the term 'Weathering' to describe mechanical, chemical, or biological influences on the earth's surface but it has to be distinguished from 'Erosion' (American Meteorological Society, 2016d). Weathering may also have an effect on climate (e.g., Köhler et al., 2010).

Common to all these usages of the root "weather" are variations, mostly referring to atmospheric processes but also their interaction with other compartments of the earth system and not just their action on these components. In the case of space weather 'variations' and 'interactions' are still considered but the 'atmospheric' component is missing. However, 'weather' may also depend on influences from space weather.

The American Meteorological Society gives the following definition of 'weather' (American Meteorological Society, 2016c):
"The state of the atmosphere, mainly with respect to its effects upon life and human activities. . . . weather consists of the short-term (minutes to days) variations in the atmosphere. . . . " The AMS's definition of an atmosphere reads (American Meteorological Society, 2016a): "A gaseous envelope gravitationally bound to a celestial body. . . " The AMS's definition of 'weather' emphasizes the minutes to days scale of the variations of an atmosphere and how life, humans, a celestial body, e.g., the planet Earth experience them.

If I want to take 'climate' as the statistics of 'weather', as noted in the introduction, I either have to take 'weather' to extend beyond an atmosphere or to refine 'climate' to refer only to atmospheres. Is the latter acceptable?

The simple answer is, that it is not acceptable. 'Climate' includes descriptions of quantities of the various compartments of the climate system (IPCC, 2013; American Meteorological Society, 2016b). If this is our first order view of 'climate', these are the statistics of 'weather' and, then, these statistics go beyond the atmosphere. Then, what is the 'weather' whose statistics
are 'climate'?

Various cited definitions of 'climate' and 'weather' use the word 'state' (e.g., American Meteorological Society, 2016c), which may be of relevance in deciding what contributes to weather. The Oxford Dictionary of English writes a state is "the particular condition that someone or something is in at a specific time" (Stevenson, 2010). Noteworthy is the wiktionary's way of putting how physics defines a state (Wiktionary, 2016): "A complete description of a system, consisting of parameters that de-
termine all properties of the system". Then the conditions of an atmosphere at a specific time are weather. Weather completely describes the 'system'. Obviously it is practically impossible to describe it completely but the concept of 'weather' encom-





passes everything that would be needed to describe the system. Does this include external, non-atmospheric dependencies or does it exclude these?

An important part of the AMS's definition of 'weather' (American Meteorological Society, 2016c) is the experience of the
variations. We experience weather instantaneous. Perceptions of weather relate to temperature, humidity, wetness, cloudiness, sunshine or no sunshine, and how windy it is. Experiences of weather also may include, e.g., its influence on the biosphere, how it influences health, and how it influences flora. Blooming of trees and flowers, budding and bursting, and pollen counts all depend on weather and are manifestations of weather. Weather influences the plankton by wind and incoming radiation and plankton influences the weather due to the flux of gases produced by plankton to the atmosphere (Kloster et al., 2006; Krüger
and Graßl, 2011).

The amount of atmospheric particulate matter is not classically seen as weather, but 'weather' may encompass it. Transport of sand or dust may be components of weather. Weather influences soil moisture which in turn may influence weather up to seasonal time-scales. Is a long heat-wave still weather? Does a flood event qualify as weather? The public may consider a flooded basement or field as weather, they may regard smog, or Sahara-dust on their cars as weather.

If one includes a set of external dependencies in the state, 'weather' extends beyond the atmosphere. Which of Earth's climate system compartments influence its atmosphere and its weather? The biosphere influences the weather, e.g., by transpiration and by its potential to emit cloud-condensation-nuclei. The cryosphere influences the weather by, e.g., the extent of its glaciers and presence and amount of sea-ice. The hydrosphere's components of oceans, surface moisture, and flow of surface-water affect the moisture-budget and the energy of the atmosphere. The lithosphere and not only the pedosphere can exert similar
influences as hydrosphere and cryosphere. Some of these processes are slow, some are fast. Which processes are reasonably seen as part of weather and which only influence our expectations of weather, i.e., climate?

The German Weather Service DWD considers the following as external to their weather forecast model (DWD, 2016a, b): ice-free ocean-surface temperature, sea-ice extent, land-sea mask, orography, surface type, albedo, land-use, vegetation-properties, and the spatial and temporal distribution of aerosols and trace gases. Here, external means they are prescribed
properties and not internally calculated. Many of these variables are slow varying aspects that provide boundary conditions on at least seasonal time-scales.

On the other hand, land-surface-temperature, land-surface liquid-water- and ice-content, the surface-snow properties, and the thickness and surface-temperature of sea-ice are internal, i.e. predicted in the DWD's forecast model (DWD, 2016a, b). These, in turn, are generally fast changing properties that directly influence and are influenced by local weather variability.
In summarising, the concept of 'weather' is applicable to bodies with atmospheres and their representations. It involves those variables determining its state at a moment in time. Quantities influencing the state can be external to the atmosphere.

There is another problem besides what counts as 'weather'. If 'weather' refers to minute to day variability, and if 'climate', as the AMS (2016b) writes, is the statistics of at least monthly averages of (earth) system variables, what is the monthly average of the (sub-)daily weather? More generally, what are time-scales of weather?





## 2.2 Time-scales of weather


If, for Earth, 'weather' takes place from minutes to days (American Meteorological Society, 2016c) and, assuming, the shortest climate time-scale is monthly (American Meteorological Society, 2016b), a conceptual definition of the transition and of the temporal difference between 'weather' and 'climate' becomes necessary. This may sound trivial but researchers and climate change activists regularly refer to interannual variability in weather and climate parameters as weather in the sense that there is not a forced response but just different weather situations from year to year. On the other hand, a multi-day mean is one simple statistic of atmospheric variations on (sub-)daily time-scales. One could see such a statistic of weather as an instance of 'climate'. Nevertheless it is usually not considered as 'climate'. The conceptual transition is not instantly clear.

The difficulty in a conceptual transition from 'weather' to 'climate' may originate from the different scientific histories of the terms. Weather is a concept of meteorology – or atmospheric physics. Climate is a descriptive concept. Climate was used to classify, weather described a dynamic evolution (compare also, Heymann, 2009, 2010).

The IPCC (2013) uses the "spatial and temporal" scale of "individual weather events" to separate climate variability from weather. This view originates in the classical scale diagrams of meteorology which plot typcial spatial against typical temporal scales (e.g., Orlanski, 1975; Daley, 1993; Cullen and Brown, 2009; Kraus, 2007; Stull, 2012; Dickinson, 1988; Norbury and Roulstone, 2002; Donner et al., 2011; Mölders and Kramm, 2014; Laing and Evans, 2011; Dolaptchiev and Klein, 2013; Klein, 2010). Insofar as these diagrams consider a climate time-scale the transition from weather to climate encompasses phenomena like the intraseasonal Madden Julian Oscillation (Zhang, 2005), the seasonal cycle, and the interannual El Niño/Southern Oscillation (Philander, 1989; Diaz and Markgraf, 2000).

One may formulate the transition dependent on the system and the process under consideration as the scale at which the predictability of an instance of 'weather' reaches its limit. The transition occurs at time-scales for which the spectra have white characteristics, where there is no memory. Indeed this is comparable to Lovejoy's (2013) view on the transition from weather to macroweather.

The transition occurs when our expectation of the properties becomes a better estimate than a deterministic forecast or as good an estimate as a probabilistic forecast. Then the upper limit of 'weather' is the time-scale at which the boundary conditions become at least as important as the initial conditions. These formulations are general and do not only apply for certain instances, e.g., weather on Earth.

Then, 'weather' are variations on a certain time-scale of variables that describe the state of a body with an atmosphere. They depend on certain aspects of a system. This system can be described by fluid dynamics and thermodynamics. The 'climate' time-scale begins and the 'weather' time-scale ends for a certain variable where our description of the system transitions from being that of a quasi-instantaneous state to that of probabilities of states over time. Related versions of this view invoke weather predictability, memory, scaling, decorrelation times, or persistence (e.g., Lorenz, unpublished, 1969a, b, 1973; Hosking, 1984; Fraedrich, 1987; Fraedrich and Ziehmann-Schlumbohm, 1994; von Storch, 1999; Baldwin et al., 2003; Fraedrich and Blender, 2003; Lovejoy, 2013). That is, the duration of the dominance of the initial conditions on this state limits the time-scales of 'weather'.





## 3 On 'climate'

If the above describes 'weather', how do we describe 'climate'? I noted that if I mean 'climate' to be the statistics of 'weather', 'weather' has to involve more than the atmosphere. This is warranted not only because the boundary conditions are relevant parts describing the state of the atmosphere, i.e. 'weather', but also because people experience phenomena involving other (earth) system compartments as weather.

The term 'climate' has been a classification of convenience from the beginning (compare Heymann, 2010; Stehr and von Storch, 2000, 1998; Barnes, 1921). It encapsulates observations of one's surroundings and helps to sort them and to compare them to other locations. It is an expression of individuals' experience and expectations of 'weather' including various earth system compartments. As such an individual's view, the roots of the classical use of the term lie more in single instances of climates than in a template of 'climate', i.e., the concept. On the other hand the simple formulation is so general that it indeed may validly describe the concept.

Bryson, Von Storch and Zwiers, the IPCC, but also others offer potentially complementary views of what 'climate' is. It is a "state" of the "climate system" (IPCC, 2013) described by "estimates of its mean ... and estimates of its variability" (von Storch and Zwiers, 2001). "[T]he thermodynamic/hydrodynamic status" of the "boundary conditions determine[s]" (Bryson, 1997) the possible occurrences of this 'state'.

### 3.1 Instances of 'climate'

It is helpful to approach a view of the template of 'climate' from instances of climate. 'Climate' in the classical sense is a description of the properties of a certain location in space and time. It allows to compare these properties between different places (e.g., Hamburg and Damaskus, or Minneapolis and Chongqing) and/or different time periods (e.g., the Middle Pleistocene and the early Holocene, or Roman times with the twentieth century) and/or different sources of data (e.g., simulations and observations, or different compilations of observations).

Climate classifications have used the meridional zones of the globe, the prevalent air masses, the aridity, or vegetation. Climate may refer to your garden, the city of Aleppo, Worcestershire, Brasil, Europe, or a celestial body with an atmosphere. The concept of 'climate' has to incorporate the climate at single locations as well as the climate at, e.g., planetary scales. It has to allow for climate at the surface, in the upper atmosphere, and in the ocean. One may want to describe the summer climate, the climate of the late 20th century, or the climate of the Pliocene. The focus may be local or global. I may consider a season in a short 20 year period or the average over millions of years. Any definition of what the term 'climate' constitutes has to serve the cultural purpose besides complying with our modern usage in studying the earth's current climate, past changes in the climate system, and future projected changes due to, e.g., anthropogenic combustion of fossil fuels.

All these are valid instances of the concept of 'climate'. No individual instance defines 'climate'. Describing instances as well as the concept requires clarity on a number of descriptors of climate like the terms 'variable', 'data', 'statistic', 'uncertainty', and 'ensembles', as well as the location and time of an instance, and the general ideas of climate forcings, climate change and climate variability.





'Climate' *variables* are potentially multiple properties or just a single property of the instance of climate in question. Climate may refer to surface temperature alone or a large set of variables like temperature, precipitation, cloudiness, incoming solar radiation, snow cover, and the seasonality of these and more variables. Variables are not necessarily basic observables but may

also derive from multiple other variables or one variable at different locations.

'Climate' *data* are not necessarily real weather observations. They may come from a model-simulation or may be inferred from proxies, i.e. from approximating representations in regions or times when there are no direct observations. Here, model does not exclusively refer to sophisticated earth system models but simply to a representation of a system that is able to reflect the behavior and evolution of the system, e.g., a time-series model. Indeed, observations are still just representations. They are

the values from a model that translates recordings by a sensor of environmental conditions into data that one can interpret in terms of weather or climate variables (compare, e.g., Evans et al., 2013).

*'Statistics'* are the results of analysing and interpreting the data (Dodge, 2006). That is, we do Statistics. We do so by calculating summary values, i.e. statistics. Each such statistic (Dodge, 2006) helps to describe the data. Common instances of statistics are expectations of the mean of a certain data or expectations of the deviation from the mean. In a first step the

interest is often an estimate of the distribution of the data, which may be achieved by estimating these expectations if the data has certain properties. In terms of climate, the interest is in the full and potentially multivariate probability distribution which likely requires other estimates beyond simple measures of mean and deviation from the mean. The statistics can become much more complex.

*Uncertainty* is important for a definition of 'climate' to be useful for research. Each data point and each statistic is an uncer-

tain, imperfect estimate. This holds even when it is not possible to quantify the uncertainty or when one only can acknowledge it implicitly. Instances of climate are unlikely perfect descriptions but uncertain approximations. This is obvious for real-world observations of properties of past climates or the current climate. Only uncertainty allows for models of climate which are by definition imprecise and uncertain. Uncertainty must be allowed for the boundary conditions, for external influences, and for the internal processes in the instance of climate. Each estimate may have a specific uncertainty. Besides these uncertain

instances, it has to be possible to derive an instance from the concept that is identical with the concept and only relies on abstract definitions.

*Ensembles* of data are common in climate research. A definition only allows for an ensemble view of different observational data sets or model simulations if it also endorses uncertainty. Then it is possible to incorporate different estimates of the same property in an instance of climate (compare, e.g., Morice et al., 2012; Donat et al., 2014; Cahill et al., 2015; Jones, 2016;

Rauser et al., 2015; Swart et al., 2015).

*Space* and *time* localise the instance, i.e., the part of a system of interest. Data describes certain properties of the system dependent on location, i.e. in all three dimensions, and time. The spatial localisation may narrowly define positions in latitude and longitude as well as in the vertical. It may mean a spatial summarizing description for a city, a county, a country, a continent, the globe. The spatial domain may be one location or an average. Local or global are the lower and upper limits of the spatial

domain.




Knowledge about the past becomes temporarily coarser resolved the further one looks back. Thus, it may be relevant to speak of the climate of the late Eocene and thereby classify climate over a time-period of millions of years, of the climate of a couple of centuries in the middle ages, or of the climate of a reference period of three well observed decades in the late 20th century. "Mean climate" may even refer to shorter periods to highlight short term variations which are thought to be not just weather but a variation or disturbance of interest of the instance of climate in question.

Such time-frames for one instance of climate may be longer or shorter dependent on the properties of the system. Climate on Mars or Jupiter may involve different time-scales. The age of the celestial body is the upper temporal limit. The lower temporal limit has to allow defining statistics and signifies the shift from a state-estimate mainly determined by its initial values, i.e. 'weather', to an uncertain expectation of the state mainly specified by its boundary conditions including further external influences.

Statistics concerning climate generally refer to estimates for a time period, i.e. our data is collected over this period. The period of time may be a collection of events rather than a consecutive sequence. The interest may be in the expected weather after a certain type of events in comparison to a climate without these events. It may be for a certain day in the year. There may also be instances when we calculate an estimate of climate over a spatial domain for one date, e.g., in the case of very low temporal resolutions of the data.

A climate is an estimate for a specific time-frame and a specific spatial scale. It varies in time and can change over time. If I change the specific period of interest I obtain another instance of climate.

### 3.1.1 The Climate system

Regarding instances of climate on our planet, variables of interest are part of the climate system or more comprehensively of the earth system. They are relevant and specific for the instance of interest at its location and on its time-scales. They describe a state which extends beyond atmospheric processes of purely meteorological interest. Summarising descriptions of the data give us the statistics of weather, e.g., the distributions of these variables. Our knowledge about these is uncertain.

Compartments of Earth's climate system are the atmosphere, the hydrosphere, the cryosphere, the biosphere, the lithosphere (IPCC, 2013). A description of the system's behavior may have to employ thermodynamics and fluid dynamics but also electromagnetism, rheology, and plasma physics besides statistics.

Then, 'climate' is a description of part of a system. It involves potentially multiple elements or just a single property of the system in question. It may include elements of any individual compartment of the system or all together as long as they are relevant. Other parts and their variations may be seen as extrinsic of the climate but interact with it.

Such external influences and external boundary conditions are *climate forcings*. In certain instances, we may consider them internal, or they may originate from internal processes. 'Climate' depends on extrinsic influences, which are also varying in time (and potentially in space).

The 'typical weather' (Bryson, 1997) or the possible weather for a certain location depends on the behavior of the system's compartments and external factors. Each compartment may depend on different external conditions but ideally it should be a common set for all.





Dependent on the specific temporal and spatial scales, the geographic specifics of the location or its surroundings differ. The climate depends on its own geographic properties and its surroundings. For example, the distribution of land, ocean, and ice-sheets conditioned a different climate for the location of the city of Hamburg in Germany 10,000 and 20,000 years ago than today's configuration. Changes in land cover or land use alter the climate. The definition of 'climate' has to allow for such influencing factors.

For specific instances such factors may be considered intrinsic parts of the climate although some may indeed be external.
For example volcanic eruptions may change the composition of the atmosphere and thus change the expectations. Processes at the surface may change the atmospheric composition. Additional 'forcings' are changes in the path on which the celestial body of interest travels around its star or variations in the composition and power of the radiation from the star or cosmic rays reaching the celestial body. The human sphere also influences the climate system on Earth.

*Climate variability* and *climate change* describe the variations of the climate. Changes are usually considered slow and refer
to detectable differences between climate statistics due to processes which we often but not necessarily may consider external to the climate system. Climate variability are faster variations within the reference time-frame which are usually thought to be intrinsic to the system and do not change the full multivariate distribution. That is, climate as the full probability distribution of certain properties in question is a distribution over the temporal variations at one or more locations dependent on extrinsic influences that are also varying and that influence the climate's evolution.

The climate is an estimate for a specific time-frame. It varies in time and can change over time with the time-frame. If I change the specific period of interest I obtain another instance of climate. Both distributions may be virtually identical or I may be able to detect that the probability of a certain event has clearly changed between both periods. Time-periods may overlap. It is important to allow for gradual changes. If statistics differ over a time period of interest from the distribution over a reference period, one speaks of change.

Regarding climate forcings, climate variability, and climate change, a slight excursion is in place. Werndl (2015) and Frigg et al. (2015) express concerns considering the robustness of classical views on climate to sudden shocks to the system. These authors define 'climate' and changes to it over different 'regimes' of external conditions.

This regime view accounts for the desired property of a definition that we can undoubtedly identify different climates. However, climate changes transiently, and defining 'climate' and its change for regimes excludes the possibility of gradual
changes. The regime view appears to only consider climate states in equilibrium. Furthermore, it is dubious, how observers can define the instance of climate directly after a sudden forcing shock.

There may not be a robust way to reconcile the wishes for a gradual definition and a definition that allows uncontroversially identifying different climates. However there may be a possible compromise. We have to assume that 'climate' also includes at least a relevant subset of forcings. If the forcing is part of the climate, e.g., volcanic eruptions, a very strong eruption can
switch 'on' the already included forcing. Even if the maximum forcing amplitude occurs quickly, the time until it reaches its maximum can be described transiently and represents a large number of infinitesimal shock events. If a climate includes all potential shock events as on-off variables, then it also accounts for the instantaneous change as the external forcing state changes to on from its off-state. If our instance of climate did not previously include the climate forcing, it seems appropriate



to assume that the spontaneous on-state of an extrinsic forcing itself changes the climate. The occurrence of the shock alters the observed system.

Processes internal to an instance of the climate system result generally only in variability of an instance of climate. Processes external to the instance of the system can change our instance of climate. However, external influences may not result in
identifiable changes, while certain internal processes may change the instance of climate under consideration. For the sake of defining 'climate', it may be necessary to include the external forcings in the instance of climate or at least consider any instance only with an associated instance of forcings. Both groups of properties vary gradually.

## 4   Discussion and Concluding remarks

The public and scientists alike often refer to climate in variations of the formulation "climate is what on an average we expect,
weather is what we actually get" (Geographical Association and London Geographical Institute, 1902). Even attempts to define the term 'climate' often fall back to more sophisticated variations on this colloquial saying (e.g., Bryson, 1997). This suggests, such a "vague" definition serves quite well the public and scientists. In turn, climate scientists may be tempted to consider the search for a rigorous definition purely philosophic.

However, regarding the policy-relevance of climate and climate change the lack of a robust definition is problematic as it at
least can lead to confusion and thus loss of time, and at the worst can prevent agreement among discussants. A robust definition aims to clarify the concept. However, policy discussions and individual views generally refer to instances. Discussants have to agree on the instance they refer to.

My thoughts above tried to defend these common views of 'climate' as the statistics of 'weather' and 'climate' as expectations of what kind of 'weather' one is going to 'experience'. My chosen approach may ask the wrong question and one cannot
conclusively define weather, climate, or at least their temporal dichotomy and terminological dependence. If that is so, the differing origins of the terms and how the view of 'climate' evolved over time (Heymann, 2009) prevent a rigorous definition. That is, in this case, there is 'weather' with its traditional definiton (e.g., American Meteorological Society, 2016c), and there is the evolving concept and understanding of 'climate' which includes weather but extends beyond it.

Then, there would be on the one hand the colloquial 20th century view of both terms which contrasts expectations and ex-
perience, i.e. 'weather' would be what we experience and 'climate' what we expect from the weather under certain conditions. As Kennedy writes on Twitter: "Practically speaking: weather's how you choose an outfit, climate's how you choose your wardrobe" (Kennedy, 2013). Both concepts would critically depend on the subjectivity of an individual or a social group and would be primarily social or cultural constructs.

On the other hand, there would be the earth-sciences' use of the terms. 'Weather' would be the concept of meteorology and
atmospheric physics, and 'climate' would be a description in terms of statistics and physics of the interplay between the various earth system compartmemts as studied by geography, climatology, biogeochemistry, oceanography, but also atmospheric sciences and others.





Conversations and discussions on climate, whether political or everyday, then have to negotiate between different constructs. In the simplest case, these are the views on climate and weather of the individuals involved. Potentially even more complex would be negotiations between the evolving scientific view on climate and such individual or social constructs. Indeed, considering the policy relevance of climate, there may be a third perspective besides the colloquial and the research-oriented, the political. This likely has additional dependencies, e.g., the economic climate, besides being culturally informed.

My thoughts above led to the following: 'Climate' is not an instantaneous property but a summary over a certain time period.
It involves the full statistical description, e.g., the distribution, of one or more properties of a system in question. The potentially multivariate distribution over the temporal variations at one or more spatial areas depends on 'extrinsic' forcings which are also varying in time and space. Any data in an instance of climate may be uncertain or may rely on different estimates of the same property. This definition holds not only for an observed instance of climate but also for a model-representation, where a model is not necessarily a complex computational simulator but may also be a simple mathematical time-series model etc.
'Climate' is gradually applicable.

Thus, can I maintain my view that 'climate' is the statistics of 'weather'? A shorter version of the above reads:

> Climate is the time-varying uncertain summarising description of multiple properties of a representation of a system for a specified spatial and temporal domain dependent on external factors varying in time and space.

If 'weather' is the description of the instantaneous time-varying state of an atmosphere and its interactions with the rest of
the system, then 'climate' is validly described as the statistics of 'weather'. However, weather has to be qualified in this sense to allow my preferred definition.

Howsoever one applies the term 'climate', it was, is, and will be an evolving concept (Heymann, 2009). It grows the larger our understanding becomes of the earth-system and of the factors influencing it and its processes, e.g., weather.

*Competing interests.* The author is not aware of any circumstances that might be seen as competing interests.

*Acknowledgements.* Comments by anonymous colleagues helped to improve the manuscript.





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
