# Peer review of "What even is 'Climate'?"

_Geoscience Communication, 2018_

## Referee Comment (RC1) · Anonymous Referee #1 · 2 Sep 2018

General This paper is an interesting run through the hugely diverse ways in which the words and concepts of weather and climate are currently used. This makes it useful to all those who encounter, or use these concepts themselves in the course of their work. It helps increase awareness of both ambiguities and incorrect uses of these terms. So in that sense it makes an interesting read, and can deepen the understanding of readers of some of the potential pitfalls of lack of clarity and consistency in the use of these terms.

However beyond the well-known and frequent confusion of 'climate' and 'weather' in the media and among lay audiences (including by climate sceptics highlighting shor-t term weather trends as invalidating long-term climate projections ) , it is not clear there is really a need for an agreed definition of 'climate' . The term is used loosely in public utterances(and has a wide semantic variation outside pure science) , but the case has not clearly been made in the paper that this variability in terminology has seriously

hindered either science or public understanding.

Even if one accepted the objective of one agreed clear definition, the definition advocated in the paper:

time-varying uncertain summarising description of multiple properties of a representation of a system for a specified spatial and temporal domain dependent on external factors varying in time and space.

seems extremely complex ( the sentence is off the scale of the FOG index) and quite incapable of comprehension outside a very limited field of scientific experts.

It might help if the authors include in the concluding section, which is where readers will want to see their 'takeaways', a plain language expansion or paraphrase of the core definition. This would be the minor revision I would suggest. The rest of the paper seems accurate and comprehensive.
* * *

---

## Referee Comment (RC2) · Anonymous Referee #2 · 27 Sep 2018

The paper analyzes various definitions of climate and weather, respectively, given by the literature. It aims at questioning these definitions and tries to provide a "new" one: "Climate is the time-varying uncertain summarising description of multiple properties of a representation of a system for a specified spatial and temporal domain dependent on external factors varying in time and space." (p. 13). However, this definition is too general and can be accounted for every "time-varying uncertain summarising description of multiple properties". Thus the proposal of a "new" definition is not convincing and even not new. It is a too general description of the main statistical definition of climate.

It doesn't become clear, why a new definition of climate is required. A good argument could be in case of policy reasons as given in the abstract. However this argument doesn't play any role in the paper anymore. Which is a pity as the author has a profound knowledge in climate research. Thus a deeper analysis of the given definitions would be helpful in developing a "better" definition of climate – "better" related to practical

reasons, e.g. for climate policy.

Strangely, the review of the definitions given by the literature stops at 2015. Contemporary literature (e.g. Heymann et al. 2017) as well as leading authors like Paul Edwards, Arthur Petersen, Wendy Parker, etc. are missing. This gives the impression as if this is an older paper.

The paper should be reworked - by providing a deeper analysis of the given definitions, - by relating the aim of developing a new definition to practical reasons, - by updating the literature, - and finally by providing concrete advice what to do with such a new definition in order to answer "what even is climate".

———————————————————

---

## Author Comment (AC1) · 8 Oct 2018

Dear referees, dear editor

Thank you for your comments on my manuscript and your ratings of my manuscript.

Below I reply to each of the referees' comments. Prior to the detailed replies, I put a summary detailing my planned changes to the manuscript.

**Summary of planned edits to the manuscript**

Assuming the editor invites a revision of my manuscript, I intend the following modifications.

[Figure]

Based on your comments, I identify three essential tasks to improve the manuscript and one additional issue.

Foremost, the manuscript requires a better foundation for its motivation, i.e., why do we need a common agreed on definition of climate. As stated below, I am uncertain, whether there really have been cases, where the lack of definition hindered policies or science. However, even if there are not any cases, I will have to provide a more robust motivation. This mainly requires edits in the introduction but possibly also modifications throughout the manuscript.

This, in turn, also requires a clearer formulation of any takeaway messages particularly in the concluding thoughts.

Thirdly, I have to re-evaluate the literature. This includes recent publications but also the full body of relevant literature on climate, climate science, and climate modelling.

Finally, referee 2 asks for a more in-depth analysis of previous definitions. Weighing the referee reports, I doubt how much the manuscript benefits from such an addition. Nevertheless, there are potentially three ways to do this. Either I simply rearrange and minimally edit the current manuscript, or, secondly, I add another subsection addressing this point. Thirdly, my favorite option is currently to mainly refer to analyses in the literature to achieve this. I am uncertain, how elaborate this will be.

**Detailed replies to the referee comments**

In the detailed replies, the referees' comments are in red font color; my replies are in normal font color.

**Referee 1**

Dear referee,

Thank you for your helpful comments on my manuscript and your ratings of the work.

General
This paper is an interesting run through the hugely diverse ways in which the words and concepts of weather and climate are currently used. This makes it useful to all those who encounter, or use these concepts themselves in the course of their work. It helps increase awareness of both ambiguities and incorrect uses of these terms. So in that sense it makes an interesting read, and can deepen the understanding of readers of some of the potential pitfalls of lack of clarity and consistency in the use of these terms.

Thank you for your kind initial evaluation.

However beyond the well-known and frequent confusion of 'climate' and 'weather' in the media and among lay audiences (including by climate sceptics highlighting short term weather trends as invalidating long-term climate projections ) , it is not clear there is really a need for an agreed definition of 'climate' . The term is used loosely in public utterances (and has a wide semantic variation outside pure science), but the case has not clearly been made in the paper that this variability in terminology has seriously hindered either science or public understanding.

There are two points here. The first is, whether we need an agreed definition? The second is that I do not make the case clearly, that the lack is problematic.

Let me first reply to the latter. You are correct, and indeed this was one of my concerns about the manuscript before submitting it. As referee 2 similarly raises this point, I will again try to identify instances where the lack of a clear definition was a substantial problem. However, I think it is possible, that there has not been a case yet where it was
clearly problematic, and it is, so far, just the nearly trivial awareness that a discussion benefits from clear statements about the discussed topic. Indeed, the requirement for an agreed on definition is a common theme in the literature on definitions of climate from Lorenz to Werndl, but as far as I can see, there is no manuscript providing case studies where the lack hindered science or political discussions on climate.

Then, the need for a common definition remains simply the opinion of authors writing on the topic.

Even if one accepted the objective of one agreed clear definition, the definition advocated in the paper:

time-varying uncertain summarising description of multiple properties of a representation of a system for a specified spatial and temporal domain dependent on external factors varying in time and space.

seems extremely complex ( the sentence is off the scale of the FOG index) and quite incapable of comprehension outside a very limited field of scientific experts.

You are correct. I have to rephrase this. Additionally, I have to try to put it better in the context.

It might help if the authors include in the concluding section, which is where readers will want to see their 'takeaways', a plain language expansion or paraphrase of the core definition. This would be the minor revision I would suggest.

I will add this.

The rest of the paper seems accurate and comprehensive.

Thank you for your encouraging comment.

Interactive
comment

**Referee 2**

Dear referee,

Thank you for your helpful comments on my manuscript and your ratings of the work.

The paper analyzes various definitions of climate and weather, respectively, given by the literature. It aims at questioning these definitions and tries to provide a "new" one:

Apparently, I am not clear enough in the scope of my manuscript. My intention is to clarify - or to defend - why the classic definitions are sufficient. Indeed, the "new" definition is more or less an accident in trying to do so.

"Climate is the time-varying uncertain summarising description of multiple properties of a representation of a system for a specified spatial and temporal domain dependent on external factors varying in time and space." (p. 13). However, this definition is too general and can be accounted for every "time-varying uncertain summarising description of multiple properties". Thus the proposal of a "new" definition is not convincing and even not new. It is a too general description of the main statistical definition of climate.

You are correct. This version as well as the classical sayings are very general. I think this generality is necessary, which motivated me to defend the classical view. Thus, my version is not meant to be new. However, this version clearly has lost its context over some revisions.

It doesn't become clear, why a new definition of climate is required. A good argument could be in case of policy reasons as given in the abstract. However this argument doesn't play any role in the paper anymore. Which is a pity as the author has a profound knowledge in climate research. Thus a deeper analysis of the given definitions would be helpful in developing a "better" definition of climate – "better" related to practical reasons, e.g. for climate policy.

You are correct. This was one of my concerns before submitting the manuscript. As
referee 1 raises a similar point, I will have to make the point better that a common/clear definition is relevant. I have to try to substantiate my points by instances where the lack of a clear definition was a substantial problem. However, I think it is possible that the lack of a definition has not clearly hindered politics or science, and that indeed so far it is just the nearly trivial awareness that a discussion benefits from clear statements about the topic discussed.

The requirement for an agreed on definition is a common theme in the literature from Lorenz to Werndl. However, as far as I can see there are apparently no manuscripts providing case studies highlighting how this had political or scientific repercussions.

Then, the need for a common definition remains simply the opinion of authors writing on the topic.

I also have to clarify that the classical statistical definition indeed has a gap in its focus on weather.

Strangely, the review of the definitions given by the literature stops at 2015. Contemporary literature (e.g. Heymann et al. 2017) as well as leading authors like Paul Edwards, Arthur Petersen, Wendy Parker, etc. are missing. This gives the impression as if this is an older paper.

Thank you for making this point as it allows me to express how much I appreciate the works of these authors.

Indeed, I did not include references to Parker, Petersen, or Winsberg. These are all authors whose contributions to the fields of the philosophy of climate science are invaluable. However, for one, they mainly concentrate on climate modelling, secondly, they do not consider the question of what climate means. Even Winsberg's book from 2018 on "Philosophy and Climate Science" does not really consider this question. Similarly, Edwards does not overly touch on this topic in his considerations on the history of climate science and climate modelling. I certainly have to re-evaluate the suggested

literature.

Further, I indeed will have to update the manuscript with respect to previous works. Shortly after publication of the manuscript, I became aware of Katzav and Parker (2018), and this led me to further older and recent publications. (Katzav and Parker, 2018: Issues in the Theoretical Foundations of Climate Science. Studies in History and Philosophy of Modern Physics 63: 141-149.)

Regarding Heymann et al. (2017): I assume you mean their book on "Cultures of Prediction". This collection of manuscripts concentrates, again, on simulations and modelling, and I regard it overall as of secondary importance for my manuscript. That is, there are more direct references for the points made by, e.g., Martin-Nielsen, Mahony, or Heymann et al. Nevertheless, the volume is a good example for the points made by, e.g., Heymann over the last decade.

Finally, the original version of the manuscript is from early 2016, but I tried to keep the references up to date. As seen above, I partially failed, but I am generally confident that the manuscript sufficiently represents the literature except for Katzav and Parker (2018), Winsberg (2018), the dissertation of Conradie (2015), and Hulme's "Weathered" (2017).

The paper should be reworked

- by providing a deeper analysis of the given definitions,

Weighing the two referee reports, the associated ratings, and the larger literature on the topic, I am, for now, uncertain, how the manuscript benefits from a deeper analysis and classification of previous definitions. To address this comment, I will try to summarise shortly the previous efforts by their main characteristics. However, this may simply result in referencing previous work.

- by relating the aim of developing a new definition to practical reasons,

As mentioned, I do not so much aim at a new definition but at showing that the classical

view is sufficient. Obviously, I have to try to clearer show the necessity of a common view on climate.

- by updating the literature,

I will reconsider the climate modelling literature as well as the literature on philosophy and history of climate science, and I have to refer to Katzav and Parker (2018).

- and finally by providing concrete advice what to do with such a new definition in order to answer "what even is climate".

I may misunderstand your comment, but from my point of view, this is part of clearly stating practical reasons for a common view on climate (see above).

————————————

Dear referees, dear editor,

Hereby I want to thank you once more for your support.

Sincerely yours,

Oliver Bothe